# Targeting of AKT1 by miR-143-3p Suppresses Epithelial-to-Mesenchymal Transition in Prostate Cancer

**DOI:** 10.3390/cells12182207

**Published:** 2023-09-05

**Authors:** Lee Armstrong, Colin E. Willoughby, Declan J. McKenna

**Affiliations:** Genomic Medicine Research Group, Ulster University, Cromore Road, Coleraine BT52 1SA, UK; armstrong-l16@ulster.ac.uk (L.A.); c.willoughby@ulster.ac.uk (C.E.W.)

**Keywords:** prostate cancer, microRNA, miR-143-3p, epithelial-to-mesenchymal transition, AKT1, biomarker

## Abstract

An altered expression of miR-143-3p has been previously reported in prostate cancer where it is purported to play a tumor suppressor role. Evidence from other cancers suggests miR-143-3p acts as an inhibitor of epithelial-to-mesenchymal transition (EMT), a key biological process required for metastasis. However, in prostate cancer the interaction between miR-143-3p and EMT-associated mechanisms remains unclear. Therefore, this paper investigated the link between miR-143-3p and EMT in prostate cancer using in vitro and in silico analyses. PCR detected that miR-143-3p expression was significantly decreased in prostate cancer cell lines compared to normal prostate cells. Bioinformatic analysis of The Cancer Genome Atlas Prostate Adenocarcinoma (TCGA PRAD) data showed a significant downregulation of miR-143-3p in prostate cancer, correlating with pathological markers of advanced disease. Functional enrichment analysis confirmed the significant association of miR-143-3p and its target genes with EMT. The EMT-linked gene AKT1 was subsequently shown to be a novel target of miR-143-3p in prostate cancer cells. The in vitro manipulation of miR-143-3p levels significantly altered the cell proliferation, clonogenicity, migration and expression of EMT-associated markers. Further TCGA PRAD analysis suggested miR-143-3p tumor expression may be a useful predictor of disease recurrence. In summary, this is the first study to report that miR-143-3p overexpression in prostate cancer may inhibit EMT by targeting AKT1. The findings suggest miR-143-3p could be a useful diagnostic and prognostic biomarker for prostate cancer.

## 1. Introduction

Prostate cancer is a multifaceted condition impacting millions of men worldwide, primarily in regions with a high human development index [1]. The process of epithelial-to-mesenchymal transition (EMT) is critical to the progression and spread of prostate cancer [2]. EMT is a cellular reprogramming process, during which epithelial cells lose their polarity and cell–cell adhesion characteristics and acquire mesenchymal properties leading to an enhanced migratory capacity, invasiveness and resistance to apoptosis [3]. The EMT process is tightly regulated by multiple factors, including several microRNAs (miRNAs) [4]. MiRNAs are small, non-coding RNA molecules that post-transcriptionally regulate gene expression by binding to the 3′ untranslated regions (3′ UTRs) of target messenger RNAs (mRNAs), leading to their degradation or translational inhibition [5]. The dysregulation of various miRNAs has been implicated in different cancers, including prostate cancer [6]. For example, previous research by our research group has demonstrated roles for miR-200c [7], miR-24 [8], miR-210 [9], miR-21 [10] and miR-182 [11] in prostate cancer, but the full contribution of other miRNAs to the disease is not fully understood. In particular, more investigation into miRNAs that are linked to the EMT process in prostate cancer is needed [12]. With this particular focus in mind, one miRNA that is an interesting candidate for further investigation is miR-143-3p.

The expression and function of miR-143-3p has been studied in various cancers (reviewed in [13]). It is generally considered to act as a tumor suppressor since its expression is downregulated in several cancer types, including ovarian [14], gastric [15], breast [16] and lung [17] cancer. Additionally, it has been firmly linked with EMT processes in bladder cancer [18,19,20], gastric cancer [21], hepatocellular cancer [22,23], colorectal cancer [24], breast cancer [25], oral squamous cell carcinoma [26], renal cancer [27] and non-small-cell lung cancer [28].

In prostate cancer, studies on miR-143-3p have also suggested it plays a tumor suppressor role [29]. Several miR-143-3p targets have been validated in prostate cancer cells and its subsequent impact on cellular behavior through different signaling pathways has been demonstrated [30,31,32,33]. Others have shown how the overexpression of miR-143-3p may be a viable strategy for enhancing prostate cancer cell chemosensitivity to drugs [34,35]. Interestingly, a link between prostate cancer risk and certain single-nucleotide polymorphisms within the miR-143 locus has also been suggested [36,37].

However, much less is known about the contribution of miR-143-3p to EMT in prostate cancer. Only two studies to date have proposed that miR-143-3p may regulate EMT in prostate cells, but neither of these examined any of its targets [38,39]. Clearly, more work is required to determine the role of miR-143-3p in EMT and its mechanistic contribution to prostate cancer through specific targets remains to be elucidated. Therefore, this study seeks to determine the role of miR-143-3p associated with EMT in prostate cancer, focusing on its expression profile, its impact on target gene expression and its influence on cellular processes. Complementary in silico analyses are used to evaluate the clinical significance of miR-143-3p in prostate cancer progression and determine its potential as a disease biomarker.

## 2. Materials and Methods

### 2.1. Cell Culture and Transfections

The cell lines were acquired from the American Type Culture Collection (ATCC Rockville, MD, USA). An in-house genotyping service authenticated the cells, and they were verified to be mycoplasma-free (InvivoGen, Toulouse, France). The cells were utilized at a low passage number ranging from 3 to 6. RWPE-1, a normal prostate epithelial cell line, was cultured in a keratinocyte growth medium, supplemented with 5 ng/mL of human recombinant epidermal growth factor and 0.05 mg/mL of bovine pituitary extract. (Life Technologies, Paisley, UK). The human prostate cancer cell lines, DU145 and PC3, were cultivated in a RPMI-1640 medium, enriched with 10% foetal bovine serum and L-glutamine (Life Technologies). The cells were maintained at 37 °C under a humidified atmosphere of 95% air and 5% CO_2_. For miRNA transfections, in a 6-well plate, 1 × 10^5^ cells were seeded per well to ensure ~80% confluency at collection. After 24 h, cells were transfected with an miR-143-3p precursor (hsa-miR-143-3p miRCURY LNA miRNA Mimic; ID YM00470035), miR-143-3p inhibitor (anti-miR-143-3p; ID YI04101614) or non-targeting control (NTC) (Negative Control A; ID YI00199006) (Qiagen, Manchester, UK) at a final concentration of 25 nM using Lipofectamine 2000 (Life Technologies). After 72 h, cells were collected for RNA or protein extraction. Pre-miR-143-3p (double-stranded RNA molecule designed to mimic endogenous mature miR-143-3p) and anti-miR-143-3p (single-stranded oligonucleotide designed to bind and inhibit endogenous miR-143-3p) are synthetic molecules designed against the mature miR-143-3p sequence UGAGAUGAAGCACUGUAGCUC.

### 2.2. Quantitative Real-Time PCR (qRT-PCR)

Total RNA was harvested from cells with themiRNeasy Tissue/Cells Advanced Mini Kit (Qiagen, Manchester, UK) according to the manufacturer’s instructions. RNA quantity and quality was confirmed by a NanoDrop™ 2000 spectrophotometer (ThermoFisher Scientific, Waltham, MA, USA). For first-strand cDNA synthesis, a Transcriptor first strand cDNA synthesis kit (Roche, Sussex, UK) was used with 500 ng of total RNA and random primers according to the manufacturer’s instructions. PCr was subsequently performed using FastStart SYBR Green Master (Roche) on a Roche LC480 LightCycler, incorporating primer sets for the following:AKT1 (fw: GGACAAGGACGGGCACATTA, rv: CGACCGCACATCATCTCGTA),CDH1 (fw: AGTGACTGATGCTGATGCCC, rv: AATGTACTGCTGCTTGGCCT),CDH2 (fw: GTGCATGAAGGACAGCCTCT, rv: TGGAAAGCTTCTCACGGCAT),ACTA2 (fw: GTTCCGCTCCTCTCTCCAAC, rv: GTGCGGACAGGAATTGAAGC),VIM (fw: GGACCAGCTAACCAACGACA, rv: AAGGTCAAGACGTGCCAGAG), ZEB1 (fw: GCTGTTTCAAGATGTTTCCTTCCA, rv: GCCTATGCTCCACTCCTTGC),ZEB2 (fw: CAACCATGAGTCCTCCCCAC, rv: GTCTGGATCGTGGCTTCTGG),FN1 (fw: TCAGCTTCCTGGCACTTCTG, rv: TCCCTGGGGATGTGACCAAT) and housekeeping gene GAPDH (fw: GACAGTCAGCCGCATCTTCT, rv: GCGCCCAATACGACCAAATC).

The expression was normalized to GAPDH and at least three independent biological replicates were performed. qRT-PCR of miR-143-3p utilized the miRCURY LNA miRNA PCR Assays system (Qiagen). 20 ng template RNA was used for the first-strand cDNA synthesis reaction. PCR took place over 40 amplification cycles and the fluorescence was monitored on the Roche LC480 LightCycler. The normalization was against housekeeping gene snRNA U6. For all qRT-PCR miRNA analysis, at least three independent biological replicates were carried out.

### 2.3. Protein Analysis

Protein was harvested using a Cell Lysis Buffer (Abcam, Cambridge, UK) with a 2% *v*/*v* Halt™ Protease Inhibitor Cocktail (ThermoFisher Scientific). Western blots were performed using Bio-Rad mini-Protean TGX Gels and the Trans-Blot^®^ Turbo Transfer System and reagents (Bio-Rad, Watford, UK). The antibodies used for blotting were rabbit-anti-AKT1, with mouse-anti-GAPDH as the loading control (both Proteintech, Manchester, UK). The membranes were blocked in 5% milk diluted in TBS-T (0.05%), followed by incubation in secondary antibody (goat anti-rabbit IgG-HRP (1:5000) or goat anti-mouse IgG-HRP (1:5000), both Proteintech). The membrane was incubated in enhanced chemiluminescent reagent (ThermoFisher Scientific) and the signal was detected on a G:BOX F3 imaging system (Syngene, Cambridge, UK). At least three biological replicates per experiment were conducted.

### 2.4. Bioassays

For the proliferation assay, transfected cells were replated at 0.01 × 10^6^ per well. After 24 h, a 10% *v*/*v* alamarBlue™ Cell Viability Reagent was added and incubated at 37 °C under a humidified atmosphere of 95% air and 5% CO_2_ for 4 h. After incubation, the absorbance (detected at 570 and 600 nm) was measured using the FLUOstar Omega plate reader (BMG LabTech, Aylesbury, UK). For the migration assay, transfected cells were replated at 0.25 × 10^6^ per well. The cells were allowed to form a confluent monolayer. A P200 pipette tip was used to create an artificial “wound”. The wounds were imaged at 0 and 72 h. The percentage of closure was calculated as (initial gap area–remaining gap area)/initial gap area) × 100. For the colony-forming assay, transfected cells were replated at 0.03 × 10^6^ per well. The cells were incubated for 72 h at 37 °C under a humidified atmosphere of 95% air and 5% CO_2_. The cells were quantified with a cell counting function on ImageJ software. The fold change was calculated relative to the NTC. 

### 2.5. Databases and Analysis

The Cancer Genome Atlas (TCGA) Prostate Adenocarcinoma (PRAD) repository data were accessed at http://portal.gdc.cancer.gov/projects (accessed on 10 January 2023). Analysis for miR-143-3p expression data and clinical parameters was carried out with The University of California Santa Cruz Xena Functional Genomics Explorer (UCSC Xena) (http://xenabrowser.net/, accessed on 12 January 2023) [40]. CancerMIRNome [41], incorporating clusterProfiler [42], was used for the identification of negative correlation targets and functional enrichment analysis (http://bioinfo.jialab-ucr.org/CancerMIRNome/, accessed on 28 July 2022). EMT-associated targets of miR-143-3p were identified using a combination of miRTarBase (http://mirtarbase.cuhk.edu.cn/, accessed on 14 February 2023) [43], the EMT gene database dbEMT2 (http://dbemt.bioinfo-minzhao.org/, accessed on 16 February 2021) [44], EMTome (accessed at http://www.emtome.org/ on 28 July 2022) [45] and Venny 2.0 (accessed at https://bioinfogp.cnb.csic.es/tools/venny/ on 28 July 2022) [46]. The protein–protein interaction (PPI) network of AKT1 was prodcued using STRING (https://string-db.org/, accessed on 24 March 2022) [47], with additional functional enrichment analysis based on Kyoto Encyclopedia of Genes and Genomes (KEGG) annotation. Additional survival analysis was carried out using Kaplan–Meier Plotter (KM-Plotter) (http://kmplot.com/analysis/, accessed on 25 June 2023) [48]. Network analyses were performed and visualized using GeneMANIA (https://genemania.org/, accessed on 21 April 2022) [49] and miRTargetLink 2.0 (http://ccbcompute.cs.uni-saarland.de/mirtargetlink2, accessed on 22 April 2022) [50].

### 2.6. Statistics

Graphs were generated using GraphPad PRISM v9. All bar graphs show the mean ± standard error of at least three biological replicates, with statistical significance assessed via a paired *t*-test. All boxplots show mean and Tukey whiskers, with the statistical significance assessed via an unpaired *t*-test with Welch’s correction or non-parametric Kruskal–Wallis one-way ANOVA with a Dunn’s Multiple Comparison Test. Pearson’s correlation was used to generate *p*-values for scatterplots, with adjustment for multiple hypothesis testing. The statistical significance for Kaplan–Meier graphs was assessed using a log-rank (Mantel–Cox) test. For multiple hypothesis correction in functional enrichment tables, the adjusted *p*-value used the Benjamini and Hochberg procedure. For all analyses, data were considered significant where * *p* < 0.05, ** *p* < 0.01, *** *p* < 0.001 or **** *p* < 0.0001. 

## 3. Results

### 3.1. Downregulation of miR-143-3p Expression Is Associated with Prostate Cancer

The qRT-PCR results demonstrated that the expression of miR-143-3p in two prostate cancer cell lines (DU145 and PC3) was significantly lower than the normal prostate cell line RWPE1 (Figure 1a). We then substantiated this result in clinical prostate samples from the TCGA PRAD patient cohort, which also showed that the expression of miR-143-3p was significantly lower in prostate cancer tumor tissue compared to normal prostate tissue (Figure 1b). Additional UCSC Xena analysis of TCGA PRAD data highlighted that a lower expression of miR-143-3p was significantly associated with clinicopathological markers of advanced prostate cancer, including the Gleason score, pathological T stage, metastatic spread and lymph node involvement (Figure 1c–f).

Functional enrichment analysis confirmed that miR-143-3p targets several gene sets that are significantly associated with prostate cancer (Table 1, Appendix A). Additional functional enrichment analysis to help identify the key biological mechanisms involved revealed several significant gene networks associated with EMT and extracellular matrix remodeling. (Table 2, Appendix A). 

### 3.2. AKT1 Is a Novel Target of miR-143-3p in Prostate Cancer

Our objective was to discover a novel target of miR-143-3p that had not yet been identified in prostate cancer. Given that miR-143-3p appears to have a tumor suppressor role and is linked with EMT in other settings, our aim was to identify a target gene known to promote EMT, which would exhibit increased expression levels in the absence of miR-143-3p. A systematic cross-referencing of EMT genes, validated targets and negative correlation targets of miR-143-3p in the TCGA PRAD data repository resulted in the identification of ERBB3, MYO6 and AKT1 as potential candidates (Appendix A). Of these, AKT1 was selected for further investigation as we had noted that miR-143-3p was significantly associated with AKT signaling pathways in several of the functional enrichment gene networks we had already identified above. We confirmed that there was sufficient predicted miRNA–target base-pairing of the AKT1-miR-143-3p duplex structure (Appendix A). We then proceeded to show that an in vitro transient overexpression of miR-143-3p consistently led to significantly decreased AKT1 levels in RWPE1, DU145 and PC3 cells, both at the mRNA and protein level (Figure 2a,b, Appendix A). We corroborated these results with analysis of TCGA PRAD, data which confirmed that the expression profiles of AKT1 and miR-143-3p displayed a significant negative correlation in these samples (Figure 2c). We showed that AKT1 expression is significantly upregulated in prostate cancer samples compared to normal prostate tissue (Figure 2d) and in tumors with a higher Gleason score (Figure 2e), trends that are the inverse of the miR-143-3p expression profile in these samples. Together, these data provide good evidence that AKT1 is targeted by miR-143-3p. This helps explain why a loss of miR-143-3p expression is detrimental in prostate cancer, since it can lead to increased AKT1 levels that subsequently promote EMT and tumor progression.

### 3.3. miR-143-3p Influences Key EMT Markers

To confirm that miR-143-3p was impacting EMT in prostate cells, a panel of several EMT-associated gene markers were measured after a transient inhibition or overexpression of miR-143-3p in RWPE1, DU145 and PC3 cells. The expressions of FN1, ZEB2, ACTA2 and CDH2, gene markers associated with the mesenchymal phenotype, were generally reduced when miR-143-3p was overexpressed, while the epithelial marker CDH1 was significantly upregulated (Figure 3).

### 3.4. miR-143-3p Alters Proliferation, Migration and Colony-Forming Capacity of Prostate Cells

Having confirmed that miR-143-3p was impacting EMT-related genes, we hypothesized that this would therefore influence cell behavior. Since the proposed target AKT1 is essential for cellular proliferation, we first assessed the effects of miR-143-3p on proliferation. The inhibition of miR-143-3p significantly increased proliferation, whereas its overexpression significantly decreased proliferation in RWPE1, DU145 and PC3 cell lines (Figure 4a). AKT1 is also implicated in the increased migratory capacity of cancer, so it was unsurprising to see that the overexpression of miR-143-3p significantly reduced migration in RWPE1, DU145 and PC3 cell lines, while the inhibition of miR-143-3p significantly increased migration in RWPE1 cell lines (Figure 4b, Appendix A). Finally, we wanted to assess the effects of miR-143-3p on the colony-forming capacity. The overexpression of miR-143-3p significantly decreased the colony-forming capacity, while the inhibition of miR-143-3p significantly increased the colony-forming capacity (Figure 4c, Appendix A).

### 3.5. Mapping the Functional Network of the miR-143-3p/AKT1 Axis

The effects on cell growth caused by the manipulation of miR-143-3p are not surprising since targeting AKT1 will impact several important cellular mechanisms. Functional enrichment analysis of AKT1 function shows its significant association with prostate cancer and various EMT-related proteins, genes and signaling pathways (Appendix A). Additionally, an overview of the bidirectional network interactions of both molecules shows there are many other targets of miR-143-3p, several of which are linked to EMT and prostate cancer, plus many other miRNAs that are known to target AKT1 (Figure 5). Altering miR-143-3p levels will therefore have a wider effect on these other targets, which will contribute to the overall effect on cells. 

### 3.6. Potential of miR-143-3p as a Biomarker of Prostate Cancer

Given the correlation between miR-143-3p and various prostate cancer clinical parameters, we hypothesized that the measurement of miR-143-3p may have clinical utility as a diagnostic and/or prognostic biomarker for the disease. The expression of miR-143-3p and clinical outcome data from the TCGA PRAD dataset were used to investigate. ROC curve analysis of the TCGA PRAD cohort demonstrates that miR-143-3p shows high potential for differentiating between tumors and normal tissue (Figure 6a). A lower expression of miR-143-3p is significantly associated with the biochemical recurrence of prostate cancer after primary treatment (Figure 6b). Similarly, a higher expression of miR-143-3p indicated a complete response to initial therapy; however, this did not meet significance (Figure 6c). For the survival analysis, the patient cohort was divided into quartiles based on their miR-143-3p expression (low < 17.99, high > 18.92 log_2_CPM). Kaplan–Meier graphs show that those in the lowest quartile of miR-143-3p expression had significantly reduced disease-free interval and progression-free interval times, compared to those in the highest quartile (Figure 6d,e). There was no significant difference between these quartiles for overall survival (Figure 6f), although the low number of deaths in this cohort is likely to be a factor in that particular analysis. Conversely, similar survival analysis for AKT1 showed that patients with a high AKT1 expression had significantly reduced disease-free interval times (Appendix A). 

Furthermore, since the loss of miR-143-3p expression has been repeatedly linked with many other cancers, it was unsurprising to see high AUC values for miR-143-3p in multiple TCGA patient cohorts, suggesting the diagnostic potential of miR-143-3p for detecting cancer (Appendix A). Similarly, the expression levels of both miR-143-3p and AKT1 in tumor tissue is significantly correlated with survival outcomes in several other TCGA patient cohorts, indicating that they could also be a useful prognostic biomarker for different cancers (Appendix A). 

## 4. Discussion

The function of miR-143-3p has been explored in relation to various cancer types and signaling pathways, but its contribution to EMT in a prostate cancer setting has not been well characterized, so this study was designed to investigate that relationship. This is the first report showing how downregulation of miR-143-3p can contribute to the development of prostate cancer through the targeting of AKT1. 

We first established that miR-143-3p expression was significantly lower in prostate cancer samples compared to normal controls, both in cell lines and clinical samples (Figure 1a,b). Further analysis of clinical samples revealed significantly lower levels of miR-143-3p were associated with more advanced cancer, as measured using clinicopathological parameters (Figure 1c–f). These data are in agreement with previous studies, which have concluded that a general reduction in miR-143-3p expression correlates with the transition from localized to metastatic prostate disease [51,52,53]. Functional enrichment analysis further validated the link between miR-143-3p and target genes associated with both prostate cancer development (Table 1) and EMT (Table 2). This all provides further evidence that the loss of miR-143-3p is detrimental in prostate cancer, as its regulation of EMT will be reduced. This agrees with existing evidence, which suggests it has a putative tumor suppressor role controlling EMT and other interconnected cellular networks (Farooqi et al., 2019). However, to date very few studies have demonstrated a functional link between miR-143-3p and EMT in prostate cancer. We therefore wanted to find an EMT-related target of miR-143-3p that had not previously been identified in this disease.

By cross-referencing gene lists from three databases, we strategically filtered possible gene targets to select for further study (Appendix A). We identified AKT1 as the most interesting candidate to investigate in more detail, since it appeared several times in our functional enrichment analyses as a gene linked to both prostate cancer and EMT. We proceeded to experimentally confirm it as a target in vitro in three cultured prostate cell lines, demonstrating in particular that miR-143-3p overexpression significantly reduces AKT1 protein and mRNA levels (Figure 2). We corroborated this in vitro evidence with analysis of TCGA PRAD data to show a significant negative correlation between miR-143-3p and AKT1, as expected if AKT1 was a target. We propose the upregulation of AKT1 gene expression observed in tumor tissue compared to normal tissue is due to the concomitant downregulation of miR-143-3p levels. Taken together, these results suggest that AKT1 is a direct target of miR-143-3p in prostate cells, as others had shown in bladder cancer [54] and colon cancer [55] cells. This is important because if the loss of miR-143-3p expression leads to an increased AKT1 expression that will impact EMT-related pathways and promote uncontrolled cell growth [56,57].

We confirmed that the manipulation of miR-143-3p levels significantly altered the expression of EMT gene markers in prostate cells (Figure 3). We then proceeded to show that altering miR-143-3p levels affected cell behavior. The inhibition of miR-143-3p significantly increased the proliferation, migration and clonogenicity of prostate cells, whereas overexpression had the opposite effect (Figure 4). Again, this helps demonstrate why the loss of miR-143-3p is deleterious in prostate cancer, since it leads to increased cell capacity for growth. With a view to potential therapeutic intervention, these results also demonstrate that the restoration of miR-143-3p levels in prostate cells, leading to a reduction in AKT1 levels, might be a viable strategy to inhibit EMT and prevent cell growth. A previous study showed that the replacement of miR-143-3p in bladder cancer cells reduced AKT1 levels, an effect that was further enhanced by simultaneously restoring miR-145-5p, a miRNA family member of miR-143-3p [54]. Similarly, the restoration of miR-143-3p levels in combination with an EGFR inhibitor reduced AKT levels and inhibited the growth of K-Ras-driven colon cancer cells [55].

However, it is obviously too simplistic to view the effects on cell activity as being wholly due to the effect of miR-143-3p on AKT1 alone. The miR-143-3p/AKT1 axis sits at the center of a wide network of interactions (Figure 5), so it is the combined effect of all these interactions that will ultimately determine the effect of miR-143-3p upon cell behavior. Previous studies in prostate cancer have also shown that miR-143-3p can inhibit cell growth through various targets, including KRAS [55], Bcl-2 [33], ERK5 [30], LIMK1 [58], HK2 [29] and KLK2 [36]. Our data suggest the miR-143-3p targeting of AKT1 exerts a similar regulatory effect in prostate cancer cells. In relation to other cancers, similar in vitro experiments have consistently shown that miR-143-3p can inhibit proliferation, invasion and migration in different cancer cell lines, including osteosarcoma [59], ovarian cancer [14], breast cancer [60], cervical cancer [61], laryngeal squamous cell carcinoma [62], leukaemia [63] and lung cancer [19]. Again, these studies all identified different targets through which these effects were mediated. Furthermore, others have reported that miR-143-3p negatively regulates the PI3K/AKT pathway in different pathological contexts [54,64,65]. Of particular note, miR-143-3p has been reported to target K-Ras [55,66] and N-Ras [67,68], key activators of the PI3K/AKT pathway and other important signaling cascades. Together, these results highlight the importance of miR-143-3p, since it can both target AKT1 directly, as we have shown in this study, and exert a wider effect on AKT signaling by targeting upstream activators of the pathway. Additionally, miR-143 interacts with other miRNAs, which will also determine its overall effect within the cell [13]. Most significantly, it is important to remember that miR-143 and miR-145 are co-transcribed from a single bicistronic unit and share regulatory elements, so it is unsurprising that they appear to act synergistically to exert a tumor suppressor effect [37,54]. There are likely other, as yet unknown, interactions of miR-143-3p with microRNAs and other non-coding RNA species.

Given the consistent downregulation of miR-143-3p in tumor tissue, we were interested in its potential as a clinically useful biomarker. In general, miRNAs are attractive candidates as biomarkers, since they are much more stably preserved than mRNA in clinical samples, including formalin-fixed paraffin-embedded (FFPE) tissues and serum, and can be readily detected using highly specific and sensitive PCR-based assays [8]. The challenge is therefore to identify the best miRNAs to use for any given disease. In this study, the data suggest that miR-143-3p expression profiling may be a useful diagnostic biomarker, since it shows significant potential to distinguish between normal and tumor tissue (Figure 6a). However, it would also be useful to have a biomarker that could predict biochemical recurrence (BCR) in prostate cancer patients. BCR is typically defined as a rise in the blood level of prostate-specific antigen after treatment with surgery or radiation, which may indicate a return of tumor growth. Here, the data showed that patients who experience biochemical recurrence following therapy have significantly lower levels of miR-143-3p (Figure 6b). Hence, knowing the miR-143-3p levels may help predict who will respond well to treatment and who will not, as well as being useful for the longitudinal monitoring of the treatment response. This is valuable to know because the potential use of miR-143-3p as a biomarker for prostate cancer has not been well studied. One study noted lower levels of miR-143 in urinary exosomes from prostate cancer patients compared to healthy controls, which would lend itself to non-invasive biomarker profiling [69]. However, another showed no difference in circulating serum levels of miR-143 between prostate cancer patients and a control group of benign prostatic hyperplasia patients [70]. Elsewhere, miR-143 was found to be elevated in semen samples from prostate cancer patients compared to healthy controls [71]. Clearly, there are discrepancies between these findings and these studies were based on small sample numbers, so more comprehensive studies are needed in order to demonstrate value for miR-143-3p as a biomarker in prostate cancer. This is worth pursuing because there is better evidence from other cancer studies that profiling miR-143-3p expression in various patient sample types has potential as a diagnostic and/or prognostic biomarker for osteosarcoma [72], gastric cancer [15], chondrosarcoma [73], bladder cancer [74], acute myeloid leukaemia [75] and colorectal cancer [76]. These studies included receiver operating characteristic (ROC) analyses, which showed the potential of miR-143-3p levels in tissue [15], serum [72,76] or plasma [15,73,75] samples to differentiate between cancer patients and healthy controls. Similarly, osteosarcoma patients with a lower miR-143-3p expression had significantly shorter overall survival than those with a higher expression, indicating its value as a prognostic biomarker. Our analyses provide evidence to warrant similar studies in prostate cancer. However, there is a fundamental requirement to have standardized approaches in studies of miRNAs as clinical biomarkers in order to robustly evaluate their potential [10].

Nevertheless, even with more thorough studies, it is unlikely that miR-143-3p on its own would possess sufficient specificity or sensitivity to serve as a useful biomarker. It is much more likely that it would be included in a multivariate panel alongside other carefully selected genomic and/or proteomic biomarkers. We have previously highlighted the superiority of multivariate panels over single biomarkers in terms of diagnostic and prognostic value for prostate cancer [77,78]. Given the interactions we have highlighted in this paper, it would be sensible for such a panel to include additional genes or proteins within the miR-143-3p/AKT1 network of interactions, particularly if the focus is on developing a panel that emphasizes EMT. In fact, current risk prediction models for prostate cancer, such as the Stockholm-3 (STHLM3) risk-based model [79], incorporate various combinations of genomic, proteomic and clinical measurements. Our findings suggest that miR-143-3p could be a valuable addition to the list of variables included in such prediction models. Given the importance of miR-143-3p in other cancers, its potential utility as a biomarker could extend to other malignancies as well. It is worth noting a recent review highlighted miR-143 as one of seven miRNAs with considerable potential as diagnostic/prognostic biomarkers in cancer [80]. A computational analysis of TCGA PRAD data also identified miR-143 in a multiomics panel for the prediction of recurrence [81].

However, a widespread acceptance of miRNAs as useful disease biomarkers will only happen if there is evidence that they can improve clinical decision making for prostate cancer patients. There is still a requirement to understand their biological function so that the most useful miRNAs, both singly and in combination, can be identified. It is also important to acknowledge that prostate tumor tissue is very heterogenous, comprising several cell types. This means profiling the expression of miR-143-3p, or any other biomarker, is not cell-specific. For example, a previous study on prostate biopsies showed miR-143 levels are significantly reduced in tumor-associated stroma, but not tumor epithelium [51]. It may be that the miR-143-3p/AKT1 relationship is more important in certain cells. Therefore, an adoption of single-cell analysis techniques and advanced proteomics may be needed to identify the cell-specific biological role of miR-143-3p. This in turn will provide valuable insights to inform more precise diagnostics and targeted therapies [82,83].

## 5. Conclusions

We have confirmed that miR-143-3p is significantly downregulated in prostate cancer and is associated with clinicopathological markers of disease progression. This is the first study to show that miR-143-3p targets AKT1 in prostate cancer cells, and we propose that the loss of miR-143-3p is implicated in driving EMT. Additional investigation is required to examine this interaction further. As a diagnostic or prognostic biomarker in prostate cancer, miR-143-3p shows promise.

## Figures and Tables

**Figure 1 cells-12-02207-f001:**
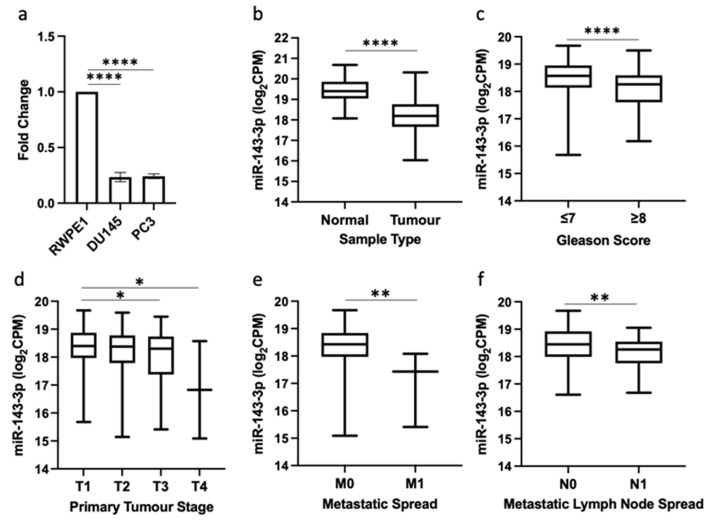
miR-143-3p is downregulated in prostate cancer and is associated with disease progression. (**a**) qRT-PCR shows miR-143-3p expression is significantly lower in DU145 and PC3 prostate cancer cell lines compared to normal prostate cell line, RWPE1 (*n =* 3, housekeeping: snRNA U6). (**b**) UCSC Xena analysis of TCGA PRAD samples shows miR-143-3p expression is significantly decreased in prostate tumor tissue (*n =* 494) compared to normal prostate tissue (*n =* 52). UCSC Xena analysis of TCGA PRAD samples shows expression of miR-143-3p is significantly lower in patients with (**c**) Gleason score ≥ 8 (*n =* 210) compared to those scored ≤ 7 (*n =* 336), (**d**) pathological stage T3 (*n =* 55) and T4 (*n =* 2) compared to T1 (*n =* 197), (**e**) pathological stage M1 (*n =* 3) compared to M0 (*n =* 542) and (**f**) pathological stage N1 (*n =* 80) compared to N0 (*n =* 385). All *p*-values generated via unpaired two-tailed *t*-test (* *p* < 0.05, ** *p* < 0.01, **** *p* < 0.0001). CPM = copies per million; *n =* number.

**Figure 2 cells-12-02207-f002:**
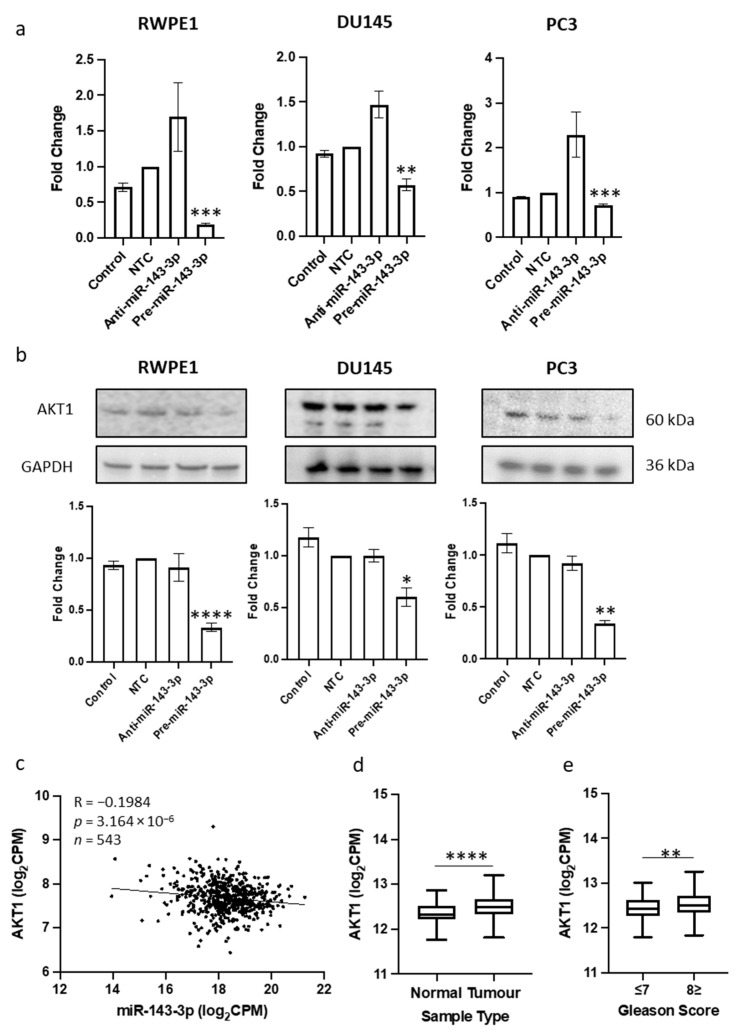
Identification of AKT1 as novel target of miR-143-3p in prostate cancer. (**a**) Transient overexpression of miR-143-3p significantly reduces AKT1 gene expression in RWPE1, DU145 and PC3 cell lines. (*n =* 3, expression values normalized to GAPDH) (**b**) Quantified Western blotting and representative images (*n =* 3) show transient overexpression of miR-143-3p causes significant downregulation of AKT1 protein in RWPE1, DU145 and PC3 cell lines. Bar graphs show mean ± SEM. *p*-values generated via unpaired two-tailed *t*-test, relative to NTC (* *p* < 0.05, ** *p* < 0.01, *** *p* < 0.001, **** *p* < 0.0001). (**c**) CancerMIRNome analysis of the TCGA PRAD specimens (*n =* 543) shows the expressions of miR-143-3p and AKT1 are significantly negatively correlated. UCSC Xena analysis of TCGA PRAD samples shows AKT1 expression is significantly elevated in (**d**) tumor tissue (*n =* 497) relative to normal (*n =* 52) tissue and in (**e**) samples with Gleason score ≥ 8 (*n* = 154) compared to those scored ≤ 7 (*n* = 213) (both Welch’s *t*-test, ** *p* < 0.01, **** *p* < 0.0001). NTC = non-targeting control; *n =* number.

**Figure 3 cells-12-02207-f003:**
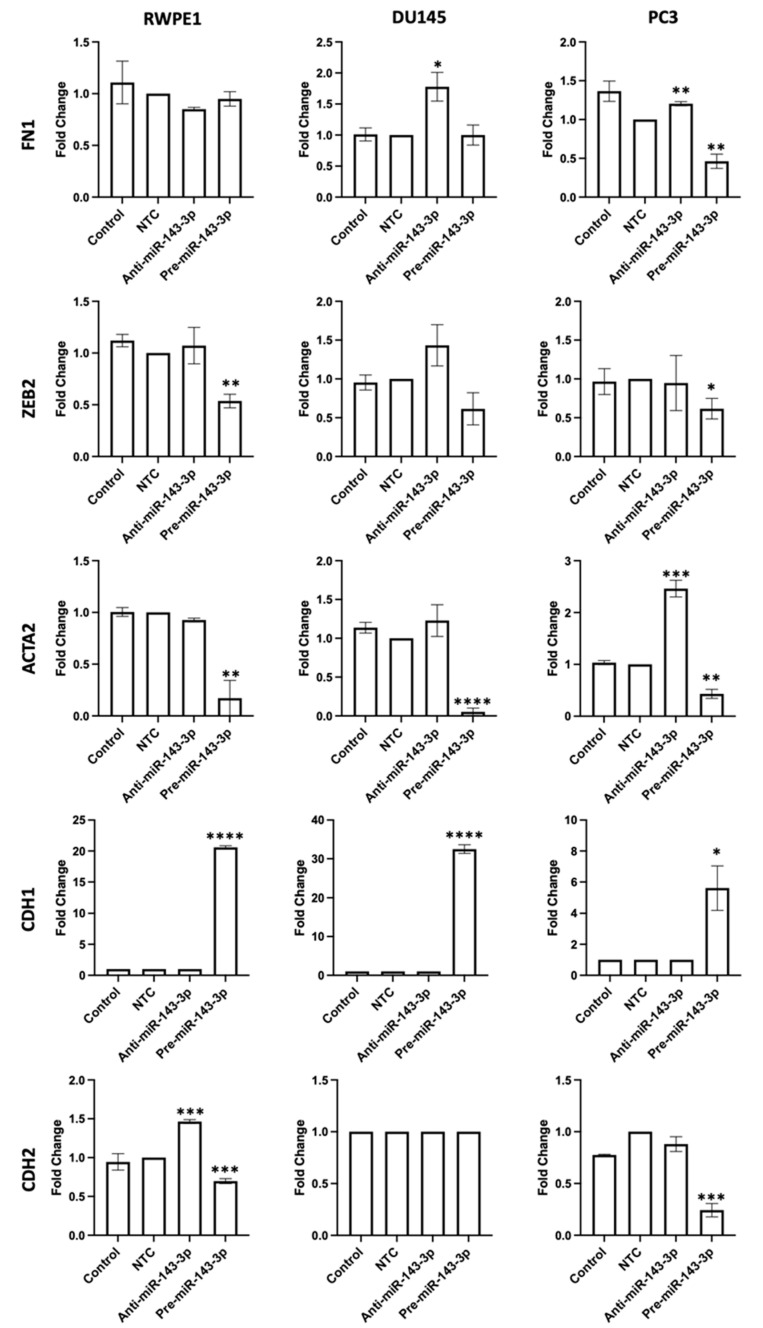
miR-143-3p alters the expression of key EMT markers. FN1, ZEB2, ACTA2, CDH1 and CDH2 quantified 72 h post-transfection using qRT-PCR. Bars show mean ± SEM (*n =* 3, expression values normalized to GAPDH). *p*-values generated via unpaired two-tailed *t*-test, relative to NTC (* *p* < 0.05, ** *p* < 0.01, *** *p* < 0.001, **** *p* < 0.0001). NTC = non-targeting control; *n =* number.

**Figure 4 cells-12-02207-f004:**
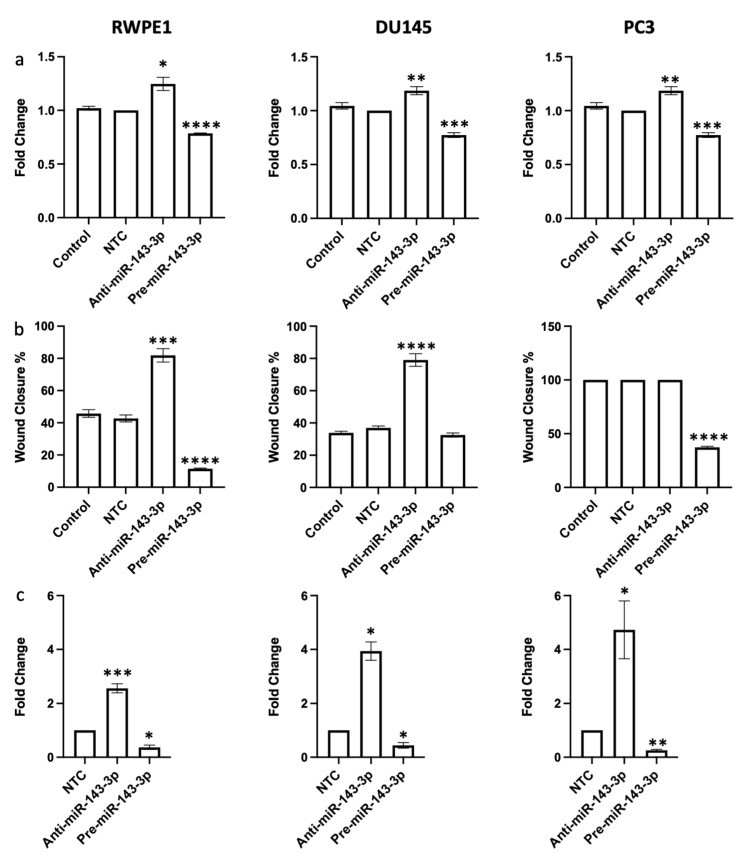
miR-143-3p alters (**a**) proliferation, (**b**) migration and (**c**) clonogenicity of RWPE1, DU145 and PC3 cells. Proliferation measured via Alamar blue absorbance (600 nm) 72 h post-transfection (*n =* 5). Migration measured via scratch assay performed at 72 h post-transfection using ImageJ to quantify wound closure (*n =* 3). Clonogenicity measured via colony counting performed at 72 h post-transfection using ImageJ to quantify. Bar graphs show mean ± SEM. *p*-values generated via unpaired two-tailed *t*-test, relative to NTC (* *p* < 0.05, ** *p* < 0.01, *** *p* < 0.001, **** *p* < 0.0001). NTC = non-targeting control; *n =* number.

**Figure 5 cells-12-02207-f005:**
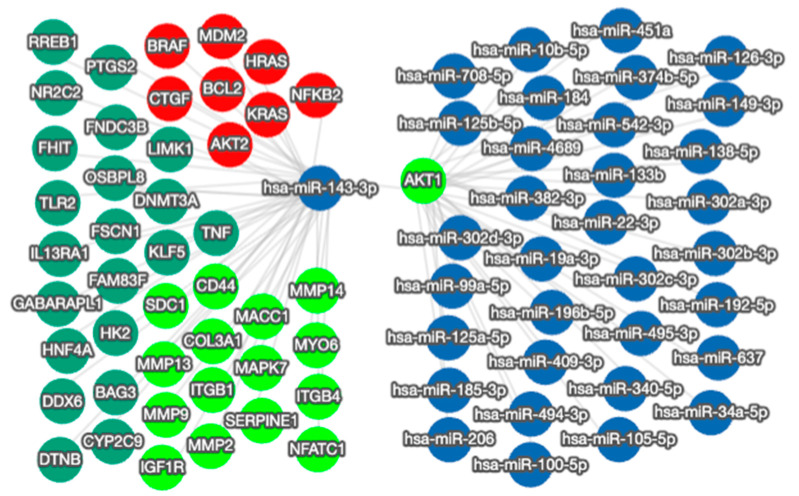
miRTargetLink 2.0 visualization of miR-143-3p and AKT1 bidirectional network interactions. miR-143-3p and AKT1 were input items, while the connected nodes are gene/miRNA interactions validated by strong experimental evidence (qRT-PCR, Western blot, Luciferase reporter assay). Blue nodes: miRNAs; green nodes: genes; bright green nodes: EMT-associated genes; red nodes: genes significantly associated with prostate cancer.

**Figure 6 cells-12-02207-f006:**
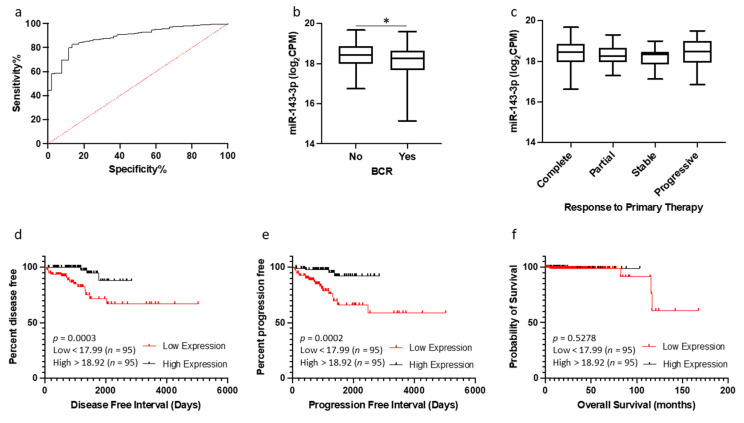
miR-143-3p as a prognostic biomarker for prostate cancer. (**a**) ROC curve analysis demonstrating that miR-143-3p shows high potential for distinguishing between tumor and normal tissue in TCGA PRAD patient cohort. (**b**) Patients experiencing biochemical recurrence (BCR) (*n* = 61) showed significantly lower levels of miR-143-3p compared to those with no recurrence (*n* = 401). *p*-value generated via unpaired two-tailed *t*-test (* *p* < 0.05). (**c**) No significant difference in miR-143-3p levels between patient remission response after primary therapy (*n*, complete = 445, partial = 41, stable = 27, progressive = 31). For KM plots, patients were divided into quartiles based on miR-143-3p expression. Quartile with lowest miR-143-3p expression showed significantly reduced time for (**d**) disease-free interval and (**e**) progression-free interval, compared to quartile with highest miR-143-3p expression. (**f**) No significant difference was found between these quartiles for overall survival. *p*-values for KM plots generated via log-rank (Mantel–Cox) test. ROC = receiver operating characteristic. AUC = area under curve. *n =* number.

**Table 1 cells-12-02207-t001:** Functional enrichment analysis of miR-143-3p in prostate cancer. Table shows the significant association of miR-143-3p target genes with gene set descriptions related to prostate cancer. Analysis performed using clusterProfiler in CancerMIRNome.

Gene Set	Gene Set ID	Description	Count/ List Total	Adjusted *p*-Value ^1^	Gene Symbol
KEGG	hsa05215	Prostate cancer	13/228	1.46 × 10^−7^	*KRAS*; *HRAS*; *AKT1*; *MDM2*; *BCL2*; *BRAF*; *MAPK1*; *PDGFRA*; *IGF1R*; *PIK3R1*; *AKT2; MMP9*; *PDGFB*
Disease Ontology	DOID:10283	Prostate cancer	25/228	1.92 × 10^−7^	*KRAS*; *MYO6*; *COL1A1*; *SERPINE1*; *FHIT*; *PTGS2*; *AKT1*; *MDM2*; *BCL2*; *SDC1*; *MAPK1*; *PDGFRA*; *SMAD3*; *CTNND1*; *IGF1R*; *TNF*; *LIMK1*; *XIAP*; *IGFBP5*; *PIK3R1*; *MMP2*; *MMP9*; *MMP14*; *ITGB1*; *ITGB4*
DisGeNET	umls:C0936223	Metastatic prostate carcinoma	13/228	1.90 × 10^−5^	*KRAS*; *PTGS2*; *JAG1*; *AKT1*; *CD44*; *CTNND1*; *ERBB3*; *TNF*; *LIMK1*; *MMP2*; *MMP9*; *MMP14*; *TERT*
	umls:C1654637	Androgen-independent prostate cancer	11/228	2.00 × 10^−5^	*COX2*; *FSCN1*; *PTGS2*; *AKT1*; *BCL2*; *MAPK1*; *TNF*; *XIAP*; *AKT2*; *MMP9*; *STAR*
	umls:C0007112	Adenocarcinoma of prostate	9/228	4.63 × 10^−4^	*KRAS*; *SERPINE1*; *FHIT*; *PTGS2*; *AKT1*; *BCL2*; *CD44*; *BRAF*; *MMP9*

KEGG = Kyoto Encyclopedia of Genes and Genomes; ^1^ Adjusted *p*-value for multiple hypothesis correction used Benjamini and Hochberg procedure.

**Table 2 cells-12-02207-t002:** Functional enrichment analysis of EMT-related functions associated with miR-143-3p. Table shows the significant association of miR-143-3p target genes with gene set descriptions related to EMT. Analysis performed using clusterProfiler in CancerMIRNome.

Gene Set	Gene Set ID	Description	Count/ List Total	Adjusted *p*-Value ^1^	Gene Symbol
KEGG	hsa04151	PI3K-Akt signaling pathway	19/228	4.15 × 10^−5^	*KRAS*; *COL1A1*; *HRAS*; *AKT1*; *MDM2*; *BCL2*; *MAPK1*; *PDGFRA*; *ERBB3*; *IGF1R*; *PPP2R5E*; *YWHAB*; *PIK3R1*; *AKT2*; *TLR2*; *ITGB1*; *ITGB4*; *PPP2R2A*; *PDGFB*
	hsa04510	Focal adhesion	14/228	4.20 × 10^−5^	*COL1A1*; *HRAS*; *AKT1*; *BCL2*; *BRAF*; *MAPK1*; *PDGFRA*; *IGF1R*; *XIAP*; *PIK3R1*; *AKT2*; *ITGB1*; *ITGB4*; *PDGFB*
	hsa04540	Gap junction	8/228	3.49 × 10^−4^	*KRAS*; *MAPK7*; *HRAS*; *MAPK1*; *PDGFRA*; *TUBB2A*; *GJD2*; *PDGFB*
REACTOME	R-HSA-3000171	Non-integrin membrane–ECM interactions	8/228	3.25 × 10^−4^	*COL1A1*; *SDC1*; *COL5A1*; *COL5A2*; *COL3A1*; *ITGB1*; *ITGB4*; *PDGFB*
	R-HSA-1474244	Extracellular matrix organization	15/228	1.86 × 10^−3^	*COL1A1*; *SERPINE1*; *MMP13*; *SDC1*; *CD44*; *COL5A1*; *COL5A2*; *COL3A1*; *ADAMTS4*; *MMP2*; *MMP9*; *MMP14*; *ITGB1*; *ITGB4*; *PDGFB*
	R-HSA-1474228	Degradation of the extracellular matrix	10/228	1.97 × 10^−3^	*COL1A1*; *MMP13*; *CD44*; *COL5A1*; *COL5A2*; *COL3A1*; *ADAMTS4*; *MMP2*; *MMP9*; *MMP14*
	R-HSA-1257604	PIP3 activates AKT signaling	12/228	8.33 × 10^−3^	*AKT1*; *MDM2*; *MAPK1*; *PDGFRA*; *ERBB3*; *PPP2R5E*; *XIAP*; *PIK3R1*; *AKT2*; *RCOR1*; *PDGFB*; *MTA3*
	R-HSA-2219528	PI3K/AKT signaling in cancer	7/228	1.04 × 10^−2^	*AKT1*; *MDM2*; *PDGFRA*; *ERBB3*; *PIK3R1*; *AKT2*; *PDGFB*
GO-BP	GO:0030198	Extracellular matrix organization	20/228	7.03 × 10^−6^	*COL1A1*; *FSCN1*; *SERPINE1*; *MMP13*; *CD44*; *COL5A1*; *PDGFRA*; *SMAD3*; *COL5A2*; *TNF*; *COL3A1*; *ADAMTS4*; *CTGF*; *MMP2*; *MMP9*; *NFKB2*; *MMP14*; *ITGB1*; *ITGB4*; *PDGFB*
	GO:0022617	Extracellular matrix disassembly	7/228	2.30 × 10^−3^	*FSCN1*; *MMP13*; *CD44*; *ADAMTS4*; *MMP2*; *MMP9*; *MMP14*
	GO:0007160	Cell–matrix adhesion	11/228	2.53 × 10^−3^	*SERPINE1*; *JAG1*; *BCL2*; *CD44*; *SMAD3*; *COL3A1*; *CTGF*; *PIK3R1*; *MMP14*; *ITGB1*; *ITGB4*
MSigDB	HEMT	HALLMARK_EPITHELIAL_MESENCHYMAL_TRANSITION	11/228	4.36 × 10^−2^	*COL1A1*; *SERPINE1*; *SDC1*; *CD44*; *COL5A1*; *COL5A2*; *COL3A1*; *CTGF*; *MMP2*; *MMP14*; *ITGB1*

KEGG = Kyoto Encyclopedia of Genes and Genomes; GO-BP = Gene Ontology-Biological Process; MSigDB:H = Molecular Signatures Database: Hallmark. ^1^ Adjusted *p*-value for multiple hypothesis correction used Benjamini and Hochberg procedure.

## Data Availability

The genotypic and phenotypic data for the Prostate Adenocarcinoma (PRAD) cohort are available at The Cancer Genome Atlas (TCGA) portal (http://portal.gdc.cancer.gov/projects) (accessed on 10 January 2023). Analysis tools are listed in the Section 2, and other datasets analyzed in the present study are available from the published papers that have been cited in this manuscript.

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
