# Peer review of "Targeting of AKT1 by miR-143-3p Suppresses Epithelial-to-Mesenchymal Transition in Prostate Cancer"

_cells, 2023, doi:10.3390/cells12182207_

Round 1

Reviewer 1 Report

The manuscript "Targeting of AKT1 by miR-143-3p suppresses Epithelial to Mesenchymal Transition in Prostate Cancer" by Lee Armstrong, Colin E. Willoughby and Declan J. McKenna is devoted to the study of the contribution of the AKT1-miR-143-3p axis to the mechanism of EMT in prostate cancer. To prove this hypothesis, they used a normal prostate epithelial cell line, the human prostate cancer cell lines DU145 and PC3, and TCGA data to evaluate expression of studied miR and its target genes in prostate tumor samples. They obtained results showing that miR-143-3p overexpression in prostate cancer may inhibit EMT by targeting AKT1. They also showed that miR-143-3p may influence on other key EMT markers. I have a question about the evidence for the interaction of miR-143-3p with AKT1, which is based on results showing that miR-143-3p significantly reduces AKT1 gene expression in RWPE1, DU145 and PC3 244 cell lines, and also using analysis of the TCGA. However, there is no data showing the theoretically possible interaction of this miR with the target to form microRNA/target duplexes, which is determined by the binding energy. This data can be found, for a example, in TargetScan. With the answer to this question, I recommend an article for publication in “Cells”.

Author Response

Thank you for the review of the paper. We appreciate the time taken by the reviewers to read our manuscript and we are pleased they recognize the potential importance and value of our work. We thank the reviewers for their insightful comments, which have helped to improve the paper.

I have a question about the evidence for the interaction of miR-143-3p with AKT1, which is based on results showing that miR-143-3p significantly reduces AKT1 gene expression in RWPE1, DU145 and PC3 244 cell lines, and also using analysis of the TCGA. However, there is no data showing the theoretically possible interaction of this miR with the target to form microRNA/target duplexes, which is determined by the binding energy. This data can be found, for a example, in TargetScan. With the answer to this question, I recommend an article for publication in “Cells”.

We provided this information in Supplementary Figure 3c, in the form of a diagram adapted from miRTarBase website, showing the computational prediction of miR-143-3p-AKT1 interaction. We believe it is best placed in supplemental material as it would otherwise clutter Figure 2, but we have now included a sentence in the results (Lines 245-247) to specifically refer to this diagram as evidence for miR-143-AKT1 complementarity

Reviewer 2 Report

The title of manuscript is remarkable. English language has good quality. There are some explainations that are needed about the section "Discussion"

1. About line 366-367 in page 14

+ why the authors have mentioned that "By cross-referencing gene lists from three databases, we identified AKT1 as the most promising candidate to investigate in more detail."?

Why the AKT1 gene is the most promising candidate?

2. About line 393-397 in page 14

Why this part has no proper reference?

3. About 413-415

The authors have mentioned that "Our data suggests that miR-143-3p expression profiling may be a useful diagnostic

biomarker and may help predict biochemical recurrence (Figure 6)"

Why you have mentioned this sentence? Please explain what were the features of miR-143-3p to be as a useful diagnostic biomarker and may help predict biochemical recurrence.

4. Please explain in detail about other surveys that have studied the diagnostic and prognostic features of miR-143-3p in the section "Discussion"

5. Please explain why the authors have not written about different connections between miR-143-3p and other types of microRNAs in prostate cancer and also the role of this connections on the functions of AKT1?

6 . Please check and adjust the "Reference

list" based on the regulations of reference list of journal. (Titles, doi, the name of journal and ... )

Round 2

Reviewer 2 Report

It is more better and now I do not have any more comment.